# Tibolone Improves Locomotor Function in a Rat Model of Spinal Cord Injury by Modulating Apoptosis and Autophagy

**DOI:** 10.3390/ijms242015285

**Published:** 2023-10-18

**Authors:** Stephanie Sánchez-Torres, Carlos Orozco-Barrios, Hermelinda Salgado-Ceballos, Julia J. Segura-Uribe, Christian Guerra-Araiza, Ángel León-Cholula, Julio Morán, Angélica Coyoy-Salgado

**Affiliations:** 1Unidad de Investigación Médica en Enfermedades Neurológicas, Hospital de Especialidades, Centro Médico Nacional Siglo XXI, Instituto Mexicano del Seguro Social, Mexico City 06720, Mexico; stephanie.sanchez.torres@gmail.com (S.S.-T.); melisalce@yahoo.com (H.S.-C.); angelleon_28@outlook.com (Á.L.-C.); 2Consejo Nacional de Ciencia y Tecnología, Mexico City 03940, Mexico; 3CONACyT-Unidad de Investigación Médica en Enfermedades Neurológicas, Hospital de Especialidades, Centro Médico Nacional Siglo XXI, Instituto Mexicano del Seguro Social, Mexico City 06720, Mexico; crls2878@gmail.com; 4Subdirección de Gestión de la Investigación, Hospital Infantil de México Federico Gómez, Secretaría de Salud, Mexico City 04530, Mexico; jujeseur@gmail.com; 5Unidad de Investigación Médica en Farmacología, Hospital de Especialidades, Centro Médico Nacional Siglo XXI, Instituto Mexicano del Seguro Social, Mexico City 06720, Mexico; christianguerra2001@gmail.com; 6División de Neurociencias, Instituto de Fisiología Celular, Universidad Nacional Autónoma de México, Mexico City 04510, Mexico; jmoran@ifc.unam.mx

**Keywords:** neuroprotection, sex hormones, cell death, motor function recovery, central nervous system, tibolone, rodent model, TUNEL assay, Western blot

## Abstract

Spinal cord injury (SCI) harms patients’ health and social and economic well-being. Unfortunately, fully effective therapeutic strategies have yet to be developed to treat this disease, affecting millions worldwide. Apoptosis and autophagy are critical cell death signaling pathways after SCI that should be targeted for early therapeutic interventions to mitigate their adverse effects and promote functional recovery. Tibolone (TIB) is a selective tissue estrogen activity regulator (STEAR) with neuroprotective properties demonstrated in some experimental models. This study aimed to investigate the effect of TIB on apoptotic cell death and autophagy after SCI and verify whether TIB promotes motor function recovery. A moderate contusion SCI was produced at thoracic level 9 (T9) in male Sprague Dawley rats. Subsequently, animals received a daily dose of TIB orally and were sacrificed at 1, 3, 14 or 30 days post-injury. Tissue samples were collected for morphometric and immunofluorescence analysis to identify tissue damage and the percentage of neurons at the injury site. Autophagic (Beclin-1, LC3-I/LC3-II, p62) and apoptotic (Caspase 3) markers were also analyzed via Western blot. Finally, motor function was assessed using the BBB scale. TIB administration significantly increased the amount of preserved tissue (*p* < 0.05), improved the recovery of motor function (*p* < 0.001) and modulated the expression of autophagy markers in a time-dependent manner while consistently inhibiting apoptosis (*p* < 0.05). Therefore, TIB could be a therapeutic alternative for the recovery of motor function after SCI.

## 1. Introduction

Spinal cord injury (SCI) results from direct damage to the spinal cord, devastating patients’ quality of life and emotional and economic stability. SCI has a worldwide incidence of approximately 10.5 per 100,000 people [1], and the average cost of treatment is around USD 2.35 million per patient [2]. Significant advances in treating SCI have been made in recent decades, including surgical decompression, hemodynamic control and methylprednisolone administration. However, these initial treatments are associated with modest functional recovery. Other treatments focusing on neuroprotective or regenerative strategies are currently under investigation, while several cellular therapies have also shown promising results. However, given that multiple factors determine the progression of SCI, a multitherapeutic approach is necessary to achieve greater efficacy in treating SCI [3,4].

SCI can be classified as primary and secondary based on the mechanism of injury [5,6]. The primary injury occurs immediately due to physical forces that cause contusion, compression, laceration or complete spinal cord segmentation [5,6]. Subsequently, secondary damage occurs when diverse biochemical events are activated, causing expansion of the area of damaged nerve tissue and aggravating neurological deficits [5,7,8].

Apoptosis and autophagy are critical in SCI [2,9]. The role of apoptosis in SCI has been demonstrated, but the exact contribution of autophagy is still controversial [10,11,12,13,14]. It has been observed that during the acute phase of SCI autophagy biomarkers increase and autophagosomes accumulate [15,16], leading to neuronal death [14,17].

Apoptotic death occurs within hours to weeks after SCI, affecting neurons, oligodendrocytes, microglia and astrocytes [18]. Caspases are the most critical mediators of apoptosis, particularly Caspase-3, which plays a crucial role in the gray and white matter of the spinal cord [19,20]. Following SCI, Caspase cleavage induces apoptotic phenotypic changes through cytoskeletal degradation, DNA fragmentation and the disruption of cellular and DNA repair processes [21]. Treatments for SCI have been investigated to develop early therapeutic interventions and mitigate the effects of apoptosis and autophagic mechanisms [17,22,23,24]. A growing body of experimental evidence has demonstrated the therapeutic benefits of estrogens (E2) in SCI [25,26,27,28], which are attributable to decreased apoptosis [26,29,30] and the regulation of autophagic mechanisms [22].

E2 treatment reduced Caspase-3 activity and neuronal death and improved locomotor function in an SCI model [26,28,30]. In another trauma SCI model, E2 decreased the expression of autophagy-related proteins such as Beclin-1 and LC3 II, improved motor function and reduced motor neuron loss by inhibiting autophagy onset [22].

Furthermore, the increased risk of breast and endometrial cancer associated with estrogen therapy has prompted the development of synthetic steroids such as tibolone (TIB) [31]. TIB is widely prescribed to treat postmenopausal symptoms and prevent bone loss [32,33].

TIB [(7α, 17α)-17-hydroxy-7-methyl-19-norpregn-5 (10) -en-20-in-3-one], a selective tissue estrogenic activity regulator, is metabolized into three biologically active metabolites: 3-α- and 3-β-hydroxy estrogenic metabolites, which bind to estrogen receptors (mainly ER alpha (ERα)), and the Δ4-keto isomer, which shows affinity for progesterone (PR) and androgen (AR) receptors [34,35]. TIB exerts tissue-specific actions depending on its local transformation [34,35,36,37]. In the central nervous system (CNS), TIB decreased neuronal death, oxidative stress and cognitive deficit in different animal models [36,38,39,40,41]. In a murine model, TIB reduced reactive gliosis in the cerebral cortex after a stab wound [42]. In astrocytes exposed to palmitic acid, TIB improved cell survival and preserved mitochondrial membrane potential [43]. Therefore, TIB is an attractive alternative to traditional estrogen therapy because of its neuroprotective actions [36,39,44,45,46,47,48,49].

On this basis, we evaluated the neuroprotective effect of TIB on neuronal death—that is, apoptosis and autophagy—and its action on the recovery of motor function in a rat model of incomplete SCI generated via moderate contusion.

## 2. Results

### 2.1. Morphometric Analysis

Fourteen days post-injury (dpi), we performed histological analysis to identify changes in spinal cord cytoarchitecture (Figure 1A). In the SCI group, we identified more microcysts contributing to the extent of damage in the rostral and caudal areas and a cellular infiltrate in the epicenter of the lesion. In contrast, in animals receiving TIB, the epicenter of the injured spinal cord was mainly occupied by tissue. TIB treatment markedly reduced the structural damage of nerve tissue in the rostral and caudal portions of the injured spinal cord compared with the untreated group. In addition, many neurons in the rostral area of the spinal cord showed preserved morphology in the TIB group. Morphometric analysis showed that nerve tissue was better preserved after SCI in animals that received TIB (Figure 1A). In addition, we performed immunohistochemistry with an antibody against NeuN as a neuronal marker to confirm whether TIB modified the number of neurons. Fourteen days after SCI, few NeuN-positive cells were observed in the rostral and caudal areas of the medullary tissue, but there were fewer in the epicenter of the lesion in the SCI group. In contrast, the NeuN marker was evident in the rostral, epicentral and caudal areas of the spinal cord in the TIB group (Figure 1B). When comparing the percentage of preserved tissue, a significant difference was observed between groups (*p* < 0.05) (Figure 1C). We also observed a significant difference between both groups in the number of NeuN-positive cells in the rostral and epicentral regions (*p* < 0.05). Although the difference in NeuN expression in the caudal area was not statistically significant, we observed a higher expression of this neuronal marker in the TIB group (Figure 1D).

### 2.2. Tibolone Administration Improves Motor Function Recovery

Recovery of motor function was assessed using the BBB scale. The SCI group obtained a final average score of 8.2 ± 0.5, meaning that most animals showed hip, knee and ankle movements at each displacement (Figure 2A). Some animals performed a sweeping gait, although without body weight support, and others showed only plantar placement of the paw without body weight support.

TIB-treated animals showed extensive movements of all three hind limb joints, with occasional body weight-supported plantar steps but no forelimb and hind limb coordination (*p* < 0.001), obtaining a score of 10.5 ± 0.3 at 30 days. Treatment with TIB allowed the animals to show faster motor recovery than those in the SCI group (Figure 2B).

### 2.3. Tibolone Regulates Autophagic Markers in a Time-Dependent Manner after Spinal Cord Injury

To determine the regulation of autophagic mechanisms with TIB treatment after SCI, we analyzed Beclin-1 and p62 protein expression and LC3-II:LC3-I ratio via Western blot assays in the laminectomy (SHAM), SCI and TIB 2.5 groups at 1, 3 and 14 dpi.

The core Beclin-1–Vacuolar protein sorting 34–Vacuolar protein sorting 15 (Vps34-Vps15) complex is required in the structure of the pre-autophagosome; hence, Beclin-1 expression correlates closely with autophagosome activity [50]. When autophagy is initiated, microtubule-associated protein 1 light chain 3 (LC3-I) undergoes ubiquitin-like changes and binds to phosphatidylethanolamine (PE) on the surface of autophagosome vacuole membranes, leading to the formation of LC3-II [51]. Therefore, the expression of LC3-II or the LC3-II:LC3-I ratio is an index reflecting the number of autophagic vacuoles [52]. Finally, p62, also known as sequestosome 1 (SQSTM1), is incorporated into autophagosomes through binding to LC3 and is degraded by the autophagic machinery. Therefore, p62 protein levels can be used to assess autophagic flux [53].

Figure 3A shows no significant changes in Beclin-1 levels at 1 and 3 dpi in the SCI group compared to the SHAM group. In contrast, Beclin-1 levels were significantly lower in the SCI group than in the SHAM group at 14 dpi. Interestingly, Beclin-1 expression in the TIB group was also significantly lower than in the SHAM and SCI groups at 3 and 14 dpi (*p* < 0.05) (Figure 3B).

In the SCI group, LC3-II:LC3-I ratio expression was significantly higher at 1 dpi, remained with no significant changes at 3 dpi and was significantly lower at 14 dpi than in the SHAM group. Conversely, TIB treatment slightly decreased the LC3-II:LC3-I ratio at 1 and 3 dpi compared with the SCI group. However, TIB significantly increased the LC3-II:LC3-I ratio at 14 dpi compared with the SCI and SHAM groups (*p* < 0.05). These results indicate that TIB decreased injury-induced autophagosome completion at 1 and 3 dpi, whereas it promoted autophagosome formation at 14 dpi (Figure 4).

In the SCI group, p62 expression was significantly lower at 1 and 3 dpi but higher at 14 dpi than in the SHAM group. In contrast, p62 expression was significantly increased at 1 dpi but decreased at 3 and 14 dpi in the TIB 2.5 group compared with the SCI group (*p* < 0.05) (Figure 5). Based on these findings, we suggest that there is a differential regulation of autophagic flux by TIB.

Overall, our results indicate that TIB promotes autophagy, even in the long term, after SCI. However, the results suggest that TIB partially decreases autophagy in the period shortly after SCI (1 and 3 dpi). Thus, these findings show that TIB modulates autophagy in a time-dependent manner.

### 2.4. Tibolone Regulates Apoptosis in Spinal Cord Injury

We evaluated cell death using the TUNEL assay and monitored active Caspase-3 expression using Western blot analysis to determine whether TIB regulated apoptosis after SCI. Figure 6 shows the mean percentage of TUNEL-positive cells in the caudal and rostral regions of the spinal cord at 14 dpi. The percentage of TUNEL-positive cells was lower in the rostral than in the caudal area (10–15%) in both groups. However, the percentage of TUNEL-positive cells in the spinal cord of TIB-treated animals was significantly lower than in the SCI group in both regions (*p* < 0.05).

Caspase-3 expression was markedly higher in the SCI group than in the SHAM group at 1, 3 and 14 dpi. In contrast, at these same dpi, TIB treatment significantly reduced Caspase-3 expression (*p* < 0.05) to levels similar to those observed in the SHAM group (Figure 7).

## 3. Discussion

In this study, we evaluated the effect of TIB on neuronal death and the recovery of motor function in a rat model of incomplete SCI. TIB administration led to a higher percentage of preserved tissue and more neurons in the rostral, epicentral and caudal areas after SCI. In addition, TIB administration improved the recovery of motor function by modulating apoptosis and autophagy.

Treatments for SCI in humans are often limited and do not allow for full recovery of lost function [3,4]. Although estrogens [22,26,54,55,56,57,58,59], progesterone [60] and selective-estrogen receptor modulators (SERMs) [61,62,63,64,65] have been shown to improve motor function scores in animal models of SCI, only a few studies have been conducted on the action of steroid hormones in humans with any SCI [66]. In addition, hormone therapy has limitations due to potential side effects [67,68,69,70]. Here, we evaluated the action of TIB in the recovery of motor function in a rat model of SCI, as the active metabolites of TIB target specific tissues, particularly in the CNS, over risk tissue (breast, endometrium) [71], and confer neuroprotection against neuronal damage [36,39,72,73].

In the present study, we observed a better recovery of motor function in TIB-treated animals after spinal cord contusion. We found a difference of 2.3 points in BBB scores between the TIB 2.5 (score 10.5) and the SCI (score 8.2) groups. A score of 10.5 indicates that animals occasionally or frequently begin to support their weight with plantar steps with no forelimb and hindlimb coordination. Conversely, a score of 8.2 means that animals only sweep without supporting body weight [74]. In 2015, Letaif et al. reported that administration of 17β-E2 (4 mg/kg) immediately after mild SCI (10 g impact rod from a standardized height of 12.5 mm) showed neuroprotective effects from the fourth week after injury and allowed recovery of motor function in these animals [27]. In contrast to Letaif et al., we generated a moderate SCI (10 g rod from a distance of 25 mm), highlighting the use of TIB to treat this type of injury. Moreover, Letaif et al. used a much higher dose of 17β-E2 (4 mg/kg) than Samantaray et al. (2016), who reported that low doses of 17β-E2 (5–10 µg) showed no significant side effects or toxicity [27,30]. Due to the undesirable effects of E2 at high doses, TIB becomes a more attractive treatment option.

We also found that functional recovery was consistent with the amount of preserved tissue, as evaluated by morphometric analysis of the spinal cord. Animals that received TIB after SCI showed more preserved tissue than those that received vehicle. This observation may be related to the estrogenic-type neuroprotective effect of TIB, as the size of injury after moderate contusion (from drop weight or impactor) to the spinal cord decreased and the amount of preserved tissue increased [75]. It is important to note that the neuroprotective effects of estrogen were primarily observed at higher doses, and no effect was observed in two studies using lower doses [58,76]. It should also be considered that, in some of these studies, treatment was administered before performing the SCI [55,63,76].

We administered TIB treatment 30 min after SCI, which was crucial for the motor recovery observed in the animals. Moreover, TIB was administered continuously, which could also be critical in maintaining its neuroprotective effect because during the intermediate chronic phases of SCI the degenerative effects of SCI persist through continued neuronal and oligodendrocyte cell death caused by injury-associated events such as Wallerian degeneration, demyelination, glial scar formation and ongoing gliosis, among others [77,78]. One study found that estrogens improved BBB scores if administered between 2 and 6 h after SCI but not if administered 12 h later [56]. Colón et al. (2016) demonstrated that immediate tamoxifen administration after spinal cord injury in rats significantly improved locomotor function compared to the improvement observed in a group treated 24 h after injury [62]. Thus, the early administration of TIB after SCI is a determinant of the observed effect on the recovery of motor function. However, further experiments will be necessary to determine the therapeutic window of TIB.

The beneficial effects of TIB may be related to its neuroprotective activity against injury-induced apoptosis, as a reduction in lesion size and functional recovery were observed after SCI. Caspase-3 correlates with apoptotic cell death after SCI [79]. The present study showed that, 14 days after injury, TIB treatment reduced Caspase-3 expression at all time points evaluated and showed a lower percentage of TUNEL-positive cells in the caudal and rostral regions as well as a higher percentage of preserved tissue compared to the SCI group. Therefore, the neuroprotective effect of TIB could be partly mediated by the modulation of Caspase-3 expression after SCI. Our results are consistent with other studies showing that estrogen therapy administered 1 to 7 dpi, as well as treatment with bazedoxifene [61] or tamoxifen [64], significantly reduced the number of TUNEL-positive cells [55,56,59,80], suggesting that the neuroprotective effect of TIB might be mediated by the inhibition of Caspase-3-dependent apoptosis. Although estradiol is known to inhibit the upstream Caspase cascade and Caspase-3 [55,81], the exact action of TIB on Caspase-3-dependent apoptosis is not yet understood. TIB has also been shown to preserve mitochondrial functions, thus attenuating cell death and oxidative damage [43]. In addition, TIB restored the expression of some proteins related to transport, translation and apoptosis. Also, TIB increased proteins related to cell survival processes [82] and reduced gliosis and neuronal loss in the cerebral cortex after acute brain injury [42]. These findings suggest that TIB is a promising neuroactive steroid for reducing apoptosis.

Several studies have focused on how activation of the PI3K/Akt [83,84,85,86,87] or ERK [84] pathway can inhibit apoptosis and improve post-SCI motor recovery. As mentioned above, TIB activates PI3K and Akt [40]. Thus, TIB could exert its beneficial effects by regulating these signaling pathways. Further research is needed to understand the impact of TIB on these signaling pathways after SCI.

In addition to apoptosis, recent studies have demonstrated the importance of autophagy modulation in SCI [22,88]. Our results showed that autophagic expression increased only on the first day post-injury; however, at 14 dpi, autophagic expression decreased in this rat model of moderate contusion SCI. These results partially agree with those reported by Zhang et al. (2014), who found that the LC3-II:LC3-I ratio in the spinal cord increased rapidly at day 3, reached its peak expression at day 7 and was significantly reduced 21 days after SCI [89]. Increased and decreased autophagic activity contributes to cell survival in the spinal cord [17,90]. By generating intracellular building blocks to purge damaged proteins, autophagy maintains vital organ functions to promote cell survival under challenging circumstances, such as nutrient-starved conditions or stress. Conversely, the excessive activation of autophagy is detrimental to cells and may contribute to programmed cell death [11,15,91]. Although autophagy has been implicated in SCI, its effect is still unclear.

Interestingly, TIB appears to modulate autophagy differentially. TIB partially inhibited Beclin-1 and the LC3-II:LC3-I ratio; however, p62 expression only increased on the first day after injury, suggesting that TIB reduced autophagic flux on this day.

Several studies have evaluated the effect of some therapies on autophagy at very early intervals [22,88]. In contrast, others have assessed autophagy over extended periods [83,90,92,93] when several cell regeneration events occur after SCI, such as remyelination and vascular and neuronal reorganization [2,78].

Fourteen days after injury, TIB reduced Beclin-1 expression, increased LC3-II:LC3-I ratio and decreased p62 expression compared to the SCI group, suggesting that TIB modifies some of the proteins involved in autophagy. The acute and subacute phases of SCI pathophysiology within the first 14 dpi are characterized by cell death, including apoptosis and autophagy [2,78,94]. Several authors have shown that different treatments post-SCI, including sex hormones, improve motor function and reduce apoptosis and autophagy in experimental models [17,22,23,26,29,30,95]. Li et al. (2017) demonstrated that inhibition of autophagy by estradiol in central cerebral ischemia positively modulated neurological deficits [96]. Research in SCI has reached a similar conclusion, as administration of insulin-like growth factor-1 (IGF-1) and estradiol post-SCI inhibited autophagy and improved motor recovery [22,88]. In the present study, we observed a significant effect of TIB on autophagic parameters 14 days after SCI—an increased number of autophagic vacuoles but a slight inhibition of autophagic flux.

TIB has a similar effect to E2, facilitating motor neuron survival and enhancing SCI recovery by inhibiting extensive autophagy [22]. However, E2 has multiple impacts on autophagy. In many cases, E2 plays a role in promoting autophagy [97]. At other times, when cellular autophagy has been stimulated by hypoxia, lipopolysaccharide, spinal cord injury or ovariectomy, E2 shows a restrictive effect on the expression of some autophagy proteins [98]. These findings highlight the importance of regulating these processes for the effective treatment of SCI. Further investigation is required to determine how TIB regulates autophagy in SCI.

The mechanisms by which TIB acts on SCI still need to be fully understood, as information is limited. In this work, the effects of TIB on SCI were explored in a novel approach. We found that TIB regulates apoptosis and autophagy, thus favoring neuroprotection and the recovery of motor function. One of the main known mechanisms by which TIB exerts its effects is the activation of ERα and ERβ [37,99]. In addition, TIB could promote neuroprotection similar to estrogen and estrogenic compounds in SCI [75]. However, no studies have examined the effects of TIB treatment after SCI. Consequently, there is a limited understanding of the networks, pathways and mechanisms that may underpin TIB-mediated neuroprotection in SCI.

One of the limitations of this work is that we did not evaluate pathways that could explain our results. However, other reports have suggested mechanisms of action of TIB during SCI. For example, PI3K/Akt and AMPK/mTOR pathways could explain the observed autophagy-related results. Some studies have shown that TIB can activate PI3K/Akt [40], and estrogens can activate the PI3K/Akt pathway in an SCI context to exert anti-apoptotic effects [55,84]. Similarly, PI3K/Akt acts as an inhibitor of autophagy in SCI when insulin is administered [88], which may explain how TIB can negatively modulate autophagy during the first 3 dpi. In addition, estrogens can activate the AMPK/mTOR pathway to promote autophagy and prevent apoptosis in osteoblasts and chondrocytes [97,100]. Upregulation of TFE3, intermittent fasting and erythropoietin administration in SCI activated the AMPK/mTOR pathway [92,93,95], promoting autophagy and improving motor recovery. Given that TIB is a STEAR with estrogen-like mechanisms of action [34,99], it is possible that the AMPK/mTOR pathway is partially responsible for the results of autophagic marker expression at 14 dpi.

## 4. Materials and Methods

### 4.1. Animals

Adult male Sprague Dawley rats weighing 250–320 g were used in this study. Animals were housed under standard conditions (12 h light/dark cycles, 22 °C) and randomly divided into three groups: SHAM (laminectomy without SCI), SCI (untreated spinal cord injury) and TIB 2.5 (SCI treated with 2.5 mg/kg of tibolone).

All surgical and experimental procedures were performed following the Regulations of the Mexican General Health Law on Research and Science [101] and the Mexican Guidelines for the Care and Handling of Animals (NOM-062-ZOO-1999) [102], with the authorization of the National Committee for Scientific Research of the Mexican Institute of Social Security (protocol number R-2021-785-011). Every effort was made to minimize animal discomfort and reduce the number of animals used.

### 4.2. Surgical Procedure

Animals were anesthetized with a mixture of xylazine (Xilasyn, PiSA, Guadalajara, Mexico) and Zoletil (Zoletil^®^100, Virbac, Carros, France) at doses of 75 and 25 mg/kg body weight, respectively, administered intramuscularly. Laminectomy was performed at thoracic vertebra 9 (T9) to produce an SCI at this level using the New York University (NYU) impactor (New York University, New York, NY, USA) [103,104]. The SCI was induced by dropping a 10 g rod from a 25 mm height [105]. The presence of a hematoma at the site of injury was verified microscopically. Subsequently, the muscle and skin were sutured in layers. An antibiotic (benzathine penicillin, PiSA, Guadalajara, Mexico) was administered intramuscularly in a single dose (1,200,000 IU), and an analgesic in the drinking water (paracetamol, 5 mL/L of water) for 5 days. The animals were placed in individual boxes in the vivarium under the described conditions. The neurogenic bladder and bowel were manually emptied daily until the animal regained sphincter control. Each animal’s surgical wound and general health status were checked daily.

### 4.3. Treatments

One tablet of TIB (Livial©, 2.5 mg, Organon, Oss, Netherlands) was dissolved in 1 mL of water, and the volume needed to administer 2.5 mg per kg of body weight was calculated (i.e., for 250 g body weight, 250 μL of TIB solution was given). Vehicle (water) or TIB was administered orally. Fifty-one rats were randomly divided into three groups. Rats in the TIB 2.5 group received an initial dose of 2.5 mg/kg of TIB orally 30 min after injury. The same dose was administered daily until euthanasia. For Western blot analysis, animals from each group (n = 9) were sacrificed at 1 (n = 3), 3 (n = 3) or 14 (n = 3) days post-injury (dpi). For immunostaining assays and the quantification of preserved tissue, animals from each group (n = 4) were treated for 14 dpi and sacrificed to collect spinal cord tissue. For motor function assessment, animals from each group (n = 8) were treated for 21 dpi and sacrificed at 30 dpi (Figure 8).

### 4.4. Tissue Collection

Once the treatments were completed, rats were euthanized according to the Mexican Official Standard (NOM-033-SAG/ZOO-2014; NOM-033-ZOO1995) regarding the humane euthanasia of animals [106].

For Western blot analyses, trained personnel euthanized the animals via decapitation using a small animal guillotine (World Precision Instruments, Inc. Sarasota, FL USA; Model DCAP-M, serial 133,708 9 K) in a room in which only one animal was placed at a time. Subsequently, spinal cord tissue, including the epicenter of the lesion plus 0.5 cm in the rostral direction and 0.5 cm in the caudal direction, was collected and preserved at 4 °C. Spinal cord tissues were homogenized with protein extraction lysis buffer (150 mM NaCl, 20 mM Tris Base, 5 mM EDTA, 10% glycerol, Nonidet P-40 (Sigma Aldrich, St. Louis, MI, USA) and Complete Protease Inhibitor (Roche)). The homogenate was centrifuged at 12,500 rpm for 30 min and the supernatant was collected. Protein concentration was determined via the Bradford method (Quick Start Bradford 1X Dye Reagent, Bio-Rad, Hercules, CA, USA).

For histological assays, rats were anesthetized with pentobarbital and perfused intracardially with phosphate-buffered saline (PBS; 137 mM NaCl, 2.7 mM KCl, 10 mM Na_2_HPO_4_, 1.8 mM KH_2_PO_4_) and 4% paraformaldehyde solution at a regulated rate of 30 mL/min using a peristaltic pump. After perfusion, a 2 cm segment of the spinal cord was removed with the epicenter of the lesion in the middle, keeping an additional 1 cm from both orientations (cephalic and caudal). Tissues were fixed in a 4% paraformaldehyde solution for 8 days, followed by dehydration with 15 min treatments using a series of graded alcohols (70%, 96% and 100% ethanol), xylol and paraffin. Tissues were then embedded in paraffin blocks in a ventral–dorsal orientation. Spinal cord tissue sections 5 µm wide were obtained using a microtome RM2125 RTS (Leica, Biosystems, Deer Park, IL, USA) and mounted on poly-L-lysine-coated slides. Slides were selected from each animal, using the epicenter of the lesion and the ependymal canal as references.

### 4.5. Morphometric Analysis

Three slides per group were rehydrated in 1 min baths of paraffin, xylol, a series of graded alcohols (100%, 95%, 85%, 70% and 50% ethanol) and water. The slides were then stained with hematoxylin-eosin and covered with Entellan (Sigma Aldrich, St. Louis, MI, USA). Panoramic images were taken with a Leica Aperio^®^ CS2 brightfield scanner (Leica, Biosystems, Deer Park, IL, USA) at 1×. In addition, images were taken at 20× and 100× magnifications in three different areas of the spinal cord: the rostral area, the epicenter of the lesion and the caudal region. Fiji software (NIH Image version 1.38×) was used to determine the total area of the spinal cord, the surface of damaged tissue and the preserved tissue from four sections per slide.

### 4.6. Immunofluorescence Analysis

The selected slides were first dehydrated in a series of graded alcohols. They were then placed in a plastic Coplin jar with 10 mM citrate buffer, pH 6, inside a pressure cooker for 20 min to recover the antigen. After this process, tissues were permeabilized with PBST (0.01 M PBS and 0.1% Triton) for 30 min and nonspecific sites were blocked with 5% horse serum in PBST in a humid chamber for 30 min. The tissues were then incubated with rabbit anti-NeuN primary antibody (1:500, No. Cat. D4G40, Cell Signaling, Danvers, MA, USA) in a humid chamber at 4 °C overnight. Subsequently, tissues were washed with PBST and incubated with Alexa Fluor^®^ 488 donkey anti-rabbit secondary IgG antibody (1:500, Molecular Probes, Eugene, OH, USA) for 2 h at room temperature in the dark. The slides were then placed in a 0.1% black Sudan B (Sigma-Aldrich, St. Louis, MI, USA) solution in 70% ethanol for 15 min, after which the excess black Sudan B was discarded with PBS. The tissues were covered with Vectashield (Vector Laboratories, CA, USA) and a coverslip. The slides were observed with the Aperio FL fluorescence scanner (Leica, Biosystems, Pleasanton, CA, USA) at 1× and magnifications were taken at 20×. Specific fluorescence was quantified in the spinal cord tissue’s rostral, epicentral and caudal areas. Cells colocalized with NeuN and DAPI were quantified manually.

### 4.7. Assessment of Functional Recovery

Recovery of motor function in the open field was assessed using the Basso, Beattie and Bresnahan (BBB) locomotion scale under a double-blind scheme. The BBB scale assesses hindlimb joint movement, plantar use of paws, weight-bearing on the limbs and forelimb and hindlimb coordination during activity. The BBB scale is a 21-point scoring scale, where 0 indicates a complete absence of limb movement and 21 indicates regular activity. The first assessment was performed 24 h after injury to verify complete hindlimb paralysis. Subsequently, functional recovery was assessed weekly for four weeks [91,92].

### 4.8. Western Blot

SDS-PAGE of the protein homogenates was performed. Subsequently, they were transferred to PVDF membranes (Millipore, Burlington, MA, USA) and blocked in 5% milk Tris-buffered saline buffer (TBS, 100 mM Trizma, 150 mM NaCl, pH 7.5). Membranes were washed three times with TTBS (0.1% Tween-20 (polysorbate) TBS) for 5 min and incubated for 24 h with the respective antibodies: anti-Caspase-3 (35, 19, 17 kDa; 1:1000; cat. 14220; Cell Signaling Technology^®^), p62 (1:1000; cat. 88588; Cell Signaling Technology^®^), Beclin-1 (1:750; cat. 3495; Cell Signaling Technology^®^) or LC3B (1:2000; cat. NB100-2220; Novus-Biologicals, Ave Centennial, CO, USA). GAPDH antibody (1:1000; cat. Sc-32233; Santa Cruz Biotechnology^®^, Dallas, TX, USA) was used as a loading control for these assays. Membranes were washed three times with TTBS for 5 min and then incubated with their corresponding secondary antibody, anti-mouse (1:10,000; cat. 115-035-003; Jackson ImmunoResearch Laboratories, Inc., West Grove, PA, USA) or anti-rabbit (1:10,000; cat. 211-032-171; Jackson ImmunoResearch Laboratories, Inc), for 1 h. Once again, the membranes were washed three times with TTBS for 5 min and then incubated for 5 min in Immobilion^®^ Crescendo Western HRP Substrate (Merck Millipore, Burlington, MA, USA). Chemiluminescence of the membranes was visualized with the Fusion Fx imaging system (Vilber Lourmat, Eberhardzell, Germany) and its corresponding Fusion^®^ software. Band area analysis was performed with Fiji software (NIH Image version 1.38×).

### 4.9. Terminal Deoxynucleotidyl Transferase dUTP Nick End Labeling (TUNEL) Assay

Three slides per group were rehydrated with 5 min baths of paraffin/xylol, xylol, a graded series of alcohol (100%, 95%, 85%, 70% and 50% ethanol) and PBS. The Abcam ab66110 Brd-Red TUNEL kit was used according to the supplier’s instructions. The slides were then counterstained with DAPI and mounted with Vectashiled^®^ (VectorLabs, Zürich Switzerland). A Nikon TI eclipse confocal microscope was used to observe two spinal cord sections per slide. Specific fluorescence was quantified in three random fields (500 × 500 μm) in the cephalic and caudal regions. Total cells and TUNEL-positive cells were quantified by analyzing the blue and red channels using Adobe Photoshop and FIJI software (NIH Image version 1.38×).

### 4.10. Statistical Analysis

Data were analyzed using GraphPad Prism version 6.0.0 for Windows (Dotmatics, GraphPad Software, San Diego, CA, USA) and IBM SPSS Statistics for Windows, version 20.0 (IBM Corp., Armonk, NY, USA) software.

BBB scores were analyzed by repeated-measures ANOVA followed by Bonferroni’s post hoc analysis and presented as means ± standard error (SE). Protein expression data were analyzed by Student’s *t*-test followed by Tukey’s post hoc analysis and presented as means ± SE. TUNEL-positive cells and preserved tissue areas were analyzed via one-way ANOVA followed by Bonferroni’s post hoc analysis and presented as means ± SE. *p*-values < 0.05 were considered statistically significant.

## 5. Conclusions

The present study showed that tibolone improves locomotor function by modulating apoptosis and autophagy in a rat model of SCI. Tibolone treatment immediately after SCI reduced structural damage to nerve tissue and effectively preserved tissue around the injury site in this rat model of moderate contusion SCI. Tibolone also reduced apoptosis and Caspase-3 activity and differentially regulated autophagy at the injury site in a time-dependent manner, significantly improving motor function. Thus, tibolone may be an alternative therapy for SCI without the side effects of estradiol.

## Figures and Tables

**Figure 1 ijms-24-15285-f001:**
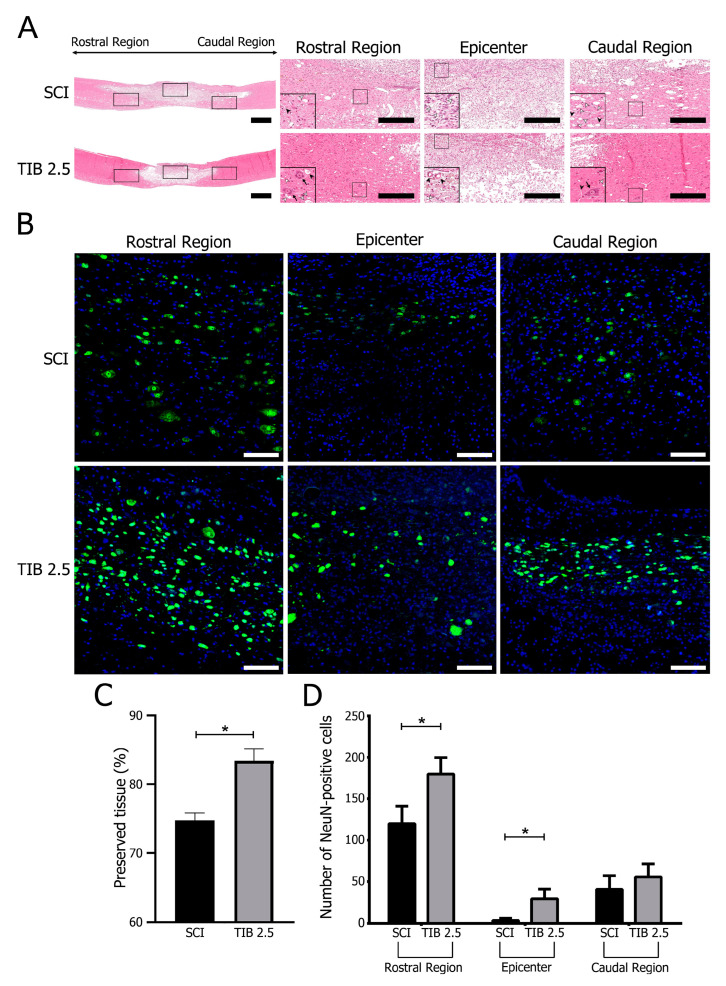
Histology and NeuN-immunoreactive cells of the spinal cord after injury. (**A**) Representative images of longitudinal spinal cord sections stained with hematoxylin-eosin 30 days after spinal cord injury. SCI, not treated; TIB 2.5, treated with tibolone (2.5 mg/kg). Images correspond to the injury’s rostral, epicentral and caudal areas at 1× (panoramic) and 20× and 100× (inset) magnifications. Blood vessels (
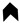
); cysts (
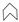
); neurons (
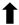
) (scale bar = 100 μm). (**B**) Representative images of immunohistochemical labeling of neurons (NeuN, green) and nuclei (DAPI, blue) in the rostral, epicentral and caudal areas of the spinal cord 14 days after injury in the SCI (n = 4) and TIB 2.5 (n = 4) groups at 20× magnification (scale bar = 100 μm). (**C**) Percentage of preserved tissue 14 days after traumatic spinal cord injury in the SCI (n = 4) or TIB 2.5 (n = 4) groups. Values are means ± SE of the percentage of preserved tissue from three histological sections per animal. Data were analyzed with one-way ANOVA followed by Bonferroni’s post hoc (* *p* < 0.05). (**D**) Comparison of the number of NeuN-positive cells colocalized with DAPI in the spinal cord’s rostral, epicentral and caudal areas. Values are means ± SE of the percentage of preserved tissue from three histological sections per animal (n = 4). A significant difference in NeuN-positive cells was observed between the SCI and TIB 2.5 groups in the rostral (Mann–Whitney’s U-test) and caudal (Student’s *t*-test) areas (* *p* < 0.05).

**Figure 2 ijms-24-15285-f002:**
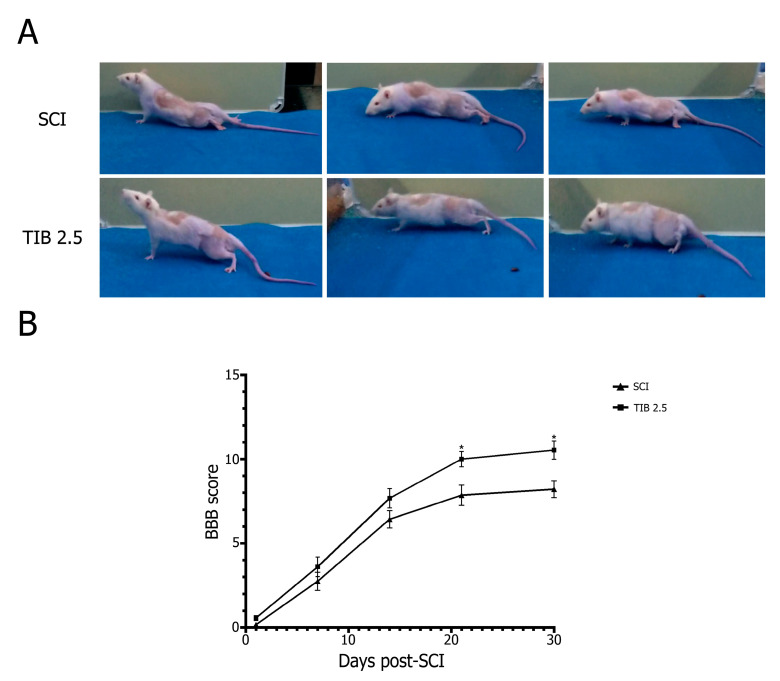
Tibolone administration promotes recovery of locomotor function. (**A**) Representative images of open-field evaluation showing recovery of locomotor function 30 days after spinal cord injury. We observed limited hind limb movements and no body weight support in the SCI group (no treatment). In contrast, extensive hind limb movements, plantar paw placement and body weight support were observed in the TIB 2.5 group. (**B**) Recovery of locomotor function was assessed in the open field with the Basso, Beattie and Bresnahan (BBB) scale, as described in the Methods section. Values are means ± SE of untreated (SCI, n = 8, triangles) or TIB-treated (TIB 2.5, n = 8, squares) animals. Data were analyzed with one-way ANOVA followed by Bonferroni’s post hoc (* *p* < 0.001).

**Figure 3 ijms-24-15285-f003:**
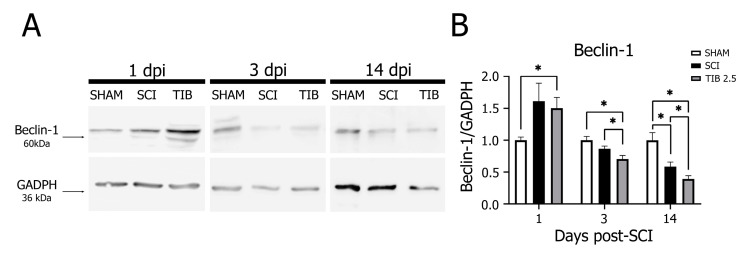
Tibolone regulates the autophagic marker Beclin-1 in a time-dependent manner after spinal cord injury. (**A**) Representative blots of Beclin-1 (~60 kDa) levels from the laminectomy (SHAM), spinal cord injury (SCI) and tibolone-treated 2.5 mg/kg (TIB 2.5) groups at 1, 3 and 14 days post-injury (dpi). GAPDH (~36 kDa) was used as a loading control. (**B**) Beclin-1 levels expressed as fold-change relative to GAPDH in the SHAM, SCI and TIB 2.5 groups at 1, 3 and 14 dpi. Values are the means ± SE of three independent determinations (n = 3). Data were analyzed with one-way ANOVA followed by Tukey’s post hoc (* *p* < 0.05).

**Figure 4 ijms-24-15285-f004:**
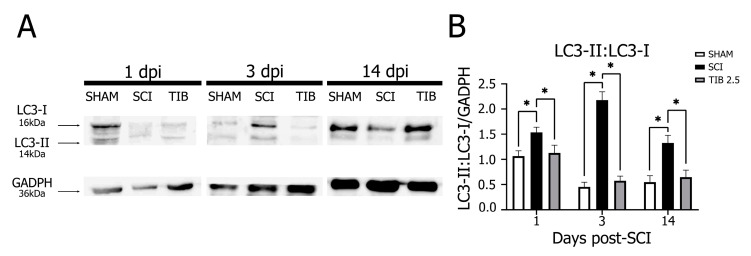
Tibolone regulates the autophagic marker LC3-II:LC3-I ratio in a time-dependent manner after spinal cord injury. (**A**) Representative blots of LC3-I (~16 kDa) and LC3-II (~14 kDa) levels from the laminectomy (SHAM), spinal cord injury (SCI) and tibolone-treated 2.5 mg/kg (TIB 2.5) groups at 1, 3 and 14 days post-injury (dpi). GAPDH (~36 kDa) was used as a loading control. (**B**) LC3-II:LC3-I ratio expressed as fold-change relative to GAPDH in the SHAM, SCI and TIB 2.5 groups at 1, 3 and 14 dpi. Values are the means ± SE of three independent determinations (n = 3). Data were analyzed with one-way ANOVA followed by Tukey’s post hoc (* *p* < 0.05).

**Figure 5 ijms-24-15285-f005:**
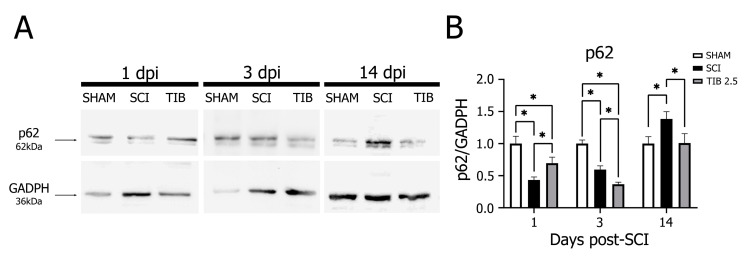
Tibolone regulates the autophagic marker p62 in a time-dependent manner after spinal cord injury. (**A**) Representative blots of p62 (~62 kDa) levels from the laminectomy (SHAM), spinal cord injury (SCI) and tibolone-treated 2.5 mg/kg (TIB 2.5) groups at 1, 3 and 14 days post-injury (dpi). GAPDH (~36 kDa) was used as a loading control. (**B**) p62 levels expressed as fold-change relative to GAPDH in the SHAM, SCI and TIB 2.5 groups at 1, 3 and 14 dpi. Values are the means ± SE of three independent determinations (n = 3). Data were analyzed with one-way ANOVA followed by Tukey’s post hoc (* *p* < 0.05).

**Figure 6 ijms-24-15285-f006:**
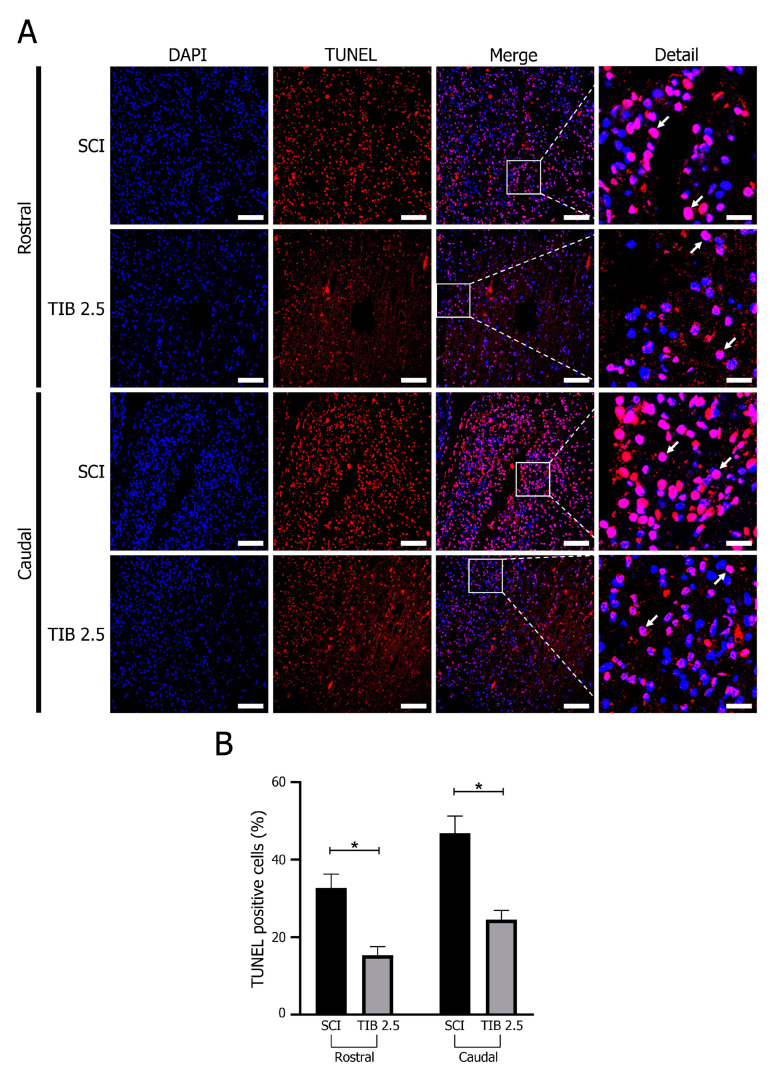
Tibolone reduced apoptosis after spinal cord injury. (**A**) Representative images of TUNEL assay 14 days post-SCI in the traumatic spinal cord injury (SCI) and tibolone-treated 2.5 mg/kg (TIB 2.5) groups in the caudal and rostral regions. Arrows indicate TUNEL-positive cells. TUNEL, terminal deoxynucleotidyl transferase dUTP nick end labeling (red), nucleus (blue). Caudal and rostral areas at 20× magnification (scale bar = 100 μm). (**B**) Percentage of TUNEL-positive cells 14 days post-injury in rostral and caudal regions. Values are means ± SE of three independent experiments (n = 3). Data were analyzed with one-way ANOVA followed by Bonferroni’s post hoc (* *p* < 0.05).

**Figure 7 ijms-24-15285-f007:**
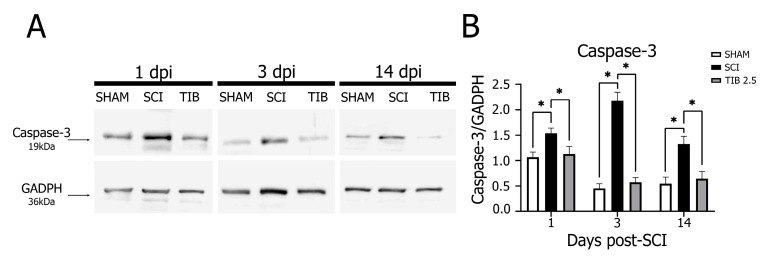
Tibolone reduced active Caspase-3 after spinal cord injury. (**A**) Representative blots of active Caspase-3 (~19 kDa) levels from the laminectomy (SHAM), spinal cord injury (SCI) and tibolone-treated 2.5 mg/kg (TIB 2.5) groups at 1, 3 and 14 days post-injury (dpi). GAPDH (~36 kDa) was used as a loading control. (**B**) Active Caspase-3 levels are expressed as fold-change relative to GAPDH in the SHAM, SCI and TIB 2.5 groups at 1, 3 and 14 dpi. Values are means ± SE of three independent experiments (n = 3). Data were analyzed with one-way ANOVA followed by Tukey’s post hoc (* *p* < 0.05).

**Figure 8 ijms-24-15285-f008:**
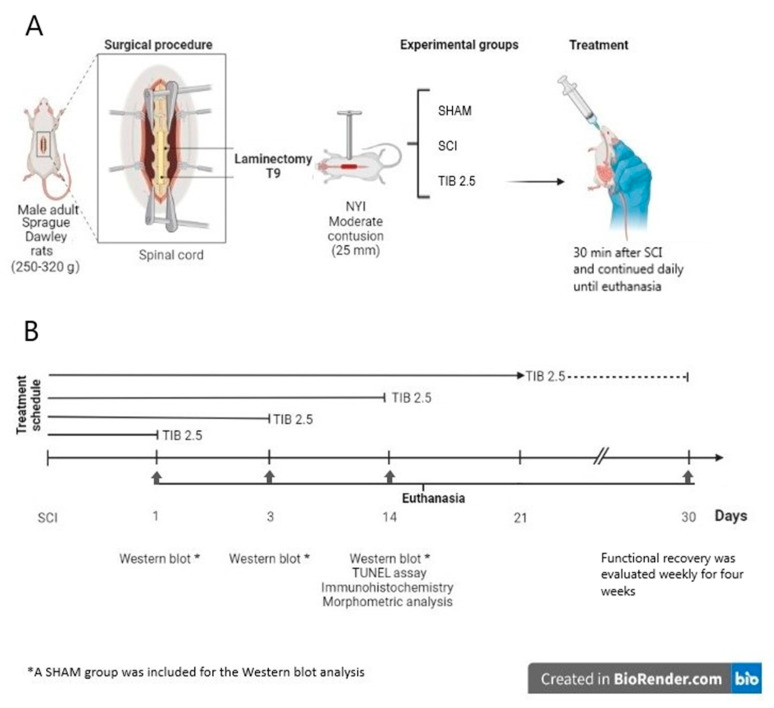
Schematic diagram of the experimental design. (**A**) Surgical procedure to perform moderate contusion spinal cord injury (SCI) at the level of thoracic vertebra 9 (T9). The animals were divided into three groups (SHAM, SCI and TIB 2.5) according to the different treatments. (**B**) Timeline of the treatments as per the analyses performed. SHAM, laminectomy; SCI, untreated; TIB 2.5, treated with tibolone 2.5 mg/kg. Created in BioRender.com (accessed on 24 July 2023).

## Data Availability

All data generated or analyzed during this study are included in this article.

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
