# Peer review of "Tibolone Improves Locomotor Function in a Rat Model of Spinal Cord Injury by Modulating Apoptosis and Autophagy"

_ijms, 2023, doi:10.3390/ijms242015285_

Round 1

Reviewer 1 Report

The manuscript titled, "Tibolone improves locomotor function in a rat spinal cord injury model by modulating apoptosis and autophagy," presents a comprehensive and in-depth investigation into the potential neuroprotective role of tibolone in mitigating neuronal death, particularly focusing on apoptosis and autophagy processes. This impact is assessed in terms of improved motor function recovery in a rat model of spinal cord injury (SCI). The authors utilize robust and suitable experimental techniques, framing their findings expertly within the context of current scientific understanding. However, for certain figures, clearer presentation and higher resolution are required to ensure a comprehensive understanding of the findings. Overall, the manuscript provides valuable insights and could contribute significantly to existing literature, subject to some revisions and clarifications.

Below are some suggestions to enhance the quality of the manuscript:

1. Introduction: The introduction provides a solid background about SCI and the role of apoptosis and autophagy in SCI. The authors have also explained the role of estrogen and the potential risks associated with its therapy leading to the selection of synthetic steroids like tibolone. The authors could provide more details about tibolone itself. Some basic information, including its mechanism of action and any previous work done using tibolone, could provide more context for the reader.

2. Results:

a) Morphometric analysis: The results are well explained. However, the description of Figure 1 could be improved. The authors should describe the visual differences observed between the SCI and TIB groups, and how these differences correlate with the stated results. It would be beneficial for both the SCI and TIB2.5 groups to provide representative images of longitudinal sections of the spinal cord “before and after” tibolone treatment. This would allow for a clear visual comparison and substantiate the reported morphometric findings. Additionally, it would be valuable to include all images of longitudinal sections of the spinal cord from the mice used in the experiment (n=4) in the supplementary data. This will ensure a more comprehensive view of the results, as it provides the opportunity for readers to observe all data points, enhancing the transparency and reliability of your findings.

b) Functional motor recovery: The functional improvement with TIB treatment is effectively demonstrated through the provided BBB scale results. A graphical representation would indeed enhance the visualization of the score differences between the SCI and TIB groups. Additionally, it would be beneficial to include immunohistochemical (IHC) results demonstrating axonal or neuronal regeneration. Specific neuronal markers can be used to confirm that neurons have regenerated, thereby providing additional evidence of the neuroprotective effect of TIB. Also, consider incorporating video evidence to visually demonstrate the difference in recovery between the two groups. This addition would provide a dynamic perspective on the beneficial effects of TIB treatment.

c). Figures 1A and 6A could benefit from more clarity and higher resolution to ensure that the reader can accurately interpret the presented data. The current images may lack the detail necessary for the reader to follow the authors' conclusions. The magnification of Figures 1A and 6A is not sufficient to discern the details of interest. The authors should consider using higher magnification in these figures to allow a more detailed examination of the cellular features. Furthermore, in addition to providing a close-up view, consider presenting an overview of the spinal cord staining in Figure 6A. This broader perspective will give readers a better understanding of the overall results.

3. In some parts, the manuscript could benefit from a clearer presentation of the statistical significance of the results. The authors should ensure the values reported in the text match those presented in the figures.

4. Language: The manuscript is well written overall. However, the authors could consider using simpler, more direct sentences for easier readability. For instance, phrases such as "based on the above" could be replaced with more precise language such as "based on these findings".

5. Discussion:

a). The authors have provided an extensive and comprehensive discussion that includes several relevant studies. However, to make the discussion easier to follow, the authors may want to consider reorganizing their arguments. They can start by explaining the results, then relating their results to existing literature, and finally offering their interpretation.

b). The paper could benefit from further elaboration on how tibolone differs from other estrogens and SERMs in its effects on the SCI model. This is briefly touched upon but could be elaborated upon to give a clearer understanding of tibolone's unique benefits.

c). The mention of potential pathways, such as PI3K/Akt and AMPK/mTOR, through which TIB may exert its effects is valuable. However, this is stated quite late in the discussion. This could be brought up earlier, and the potential roles of these pathways could be discussed more extensively.

d). In the part discussing apoptosis and autophagy, the authors may consider integrating more on the interplay between the two and how this can affect the overall outcome of SCI.

e). The authors have mentioned future research directions at the end, which is important. However, this could be expanded. Specifically, the authors might mention the potential clinical implications and the steps needed to translate these findings to clinical practice.

f). The authors could better emphasize the significance of their findings, specifically the time-dependent regulation of autophagy by TIB, in the broader context of SCI treatment.

g). Some statements, such as the potential protection of other cells like oligodendrocytes and astrocytes by TIB, are speculative. It would be better to discuss this in the future directions section, as a potential area to be explored, rather than in the results discussion.

h). Minor point: Be careful with phrasing that attributes intentionality to molecular compounds, such as "TIB demonstrated an ability to modulate autophagy." This can be rephrased to say that "TIB appears to modulate autophagy," which is more accurate since chemicals don't demonstrate intentions.

6. References: It would be helpful for the authors to make sure their references are up to date and cover the most recent advances in the field. This would strengthen their arguments and support their conclusions.

Minor editing of English language required

Reviewer 2 Report

The authors presented the results of the study on an application of Tibolone (TIB) as a selective tissue estrogen activity regulator (STEAR), demonstrating its neuroprotective properties in the rat’s model. They state that TIB increased the amount of preserved tissue, improved motor function recovery, and modulated the expression of autophagy markers in a time-dependent manner while consistently inhibiting apoptosis. They conclude that TIB could be a therapeutic alternative for recovering motor function after SCI.

The study results are convincing.

The manuscript has few flaws that should be corrected or improved before accepting the research report for printing.

1. The Abstract section does not contain the short presentation of the most important results and is actually not very informative, nor the research methods are listed, there is no data on the subjects and their number.

2. Keywords should include "Tibolon" and "rodent experimental study" and the names of the most important research methods.

3. If the authors present data on SCI in humans in an interesting and concise Introduction, they should also present the most important and effective methods of treatment together with examples of references on this topic (which, by the way, are included in the References list) before embarking on animal research. One has the impression that, in the opinion of the authors, none of the methods of pharmacological treatment, cell transplantation, rehabilitation (kinesiotherapy and physicotherapy by means of electrotherapy or rTMS) are effective. Moreover, they mention the role of Tibolone in just two paragraphs and this issue should be expanded on whether it is treated more extensively in the Discussion or not.

4. The "moderate contusion" (line 76) in aim at the end of the Introduction is perhaps better associated with "incomplete spinal cord injury" (iSCI).

5. Presenting a one-sentence post-aim conclusion (lines 77-79) is unusual, more suitable to the Conclusions section.

6. The study design should be presented in a flow chart to be clearly understood.

7. The results section is excellently presented, but it would be even more convincing when  providing examples of a microscopic photograph of an injured spinal cord cross-sections (1:1 scale iSCI), e.g. in Figure 1.

8. The sentences in lines 277-290 match the "Study limitations" which are missing the Discussion, despite its rather exhaustive content.

9. In the M&M section, NYU device should be presented wider and it's short fully explained.

10. The approach of BBB evaluation scale and the whole methodology of locomotor evaluations should be developed and presented in photographs.

11. Conclusions section is laconic, it does not contain research methods with the help of which the results presented in one or two sentences lead to specific conclusions.

Minor English editing corrections are necessary 

Reviewer 3 Report

we read the article by Heredia-Nieto entitled Tibolone improves locomotor function in a rat spinal cord injury model by modulating apoptosis and Autophagy.

the aim of the study was to evaluate the role of TIB at 2.5 mg/kg and evaluate spinal Neurotherapy at  1-, 3-, 5-, 7- or 14 days post-injury (dpi) using behavioral assessment and HE staining and immunoblotting analysis with caspase 3 , beclin, LC-3 and p63 

Major Comments:

the authors have a strange experimental design that needs to be clarified, they don't mention how many animals were used in each group (for behavioral and for the Western study), so a schematic is needed. They mention that animals were treated for 21 dpi however they were sacrificed on day 14???

the work mentioned the Laminectomy group, doe this group represents a control group not treated with anything, this is a very confusing study.

the Western blot data need to be presented as raw data, many cropped bands are showing that do not reflect the true blot, each blot should be shown without any cropping and the triplicate of each band should be shown.

the authors mention that they sacrificed animals at 1-, 3-, 5-, 7- or 14, but all data are only showing 1,3, and 14 days.

the immunofluorescence data should be repeated for other days and should be done with NeuN, and other inflammatory markers such as GFAP and Iba-1 markers to indicate inflammatory response.

the HE is not showing true morphological changes and the authors mention that the neurons in the rostral zone of the spinal cord showed a conserved morphology. this is a weird sentence that I can't interpret. specific neuronal markers should be performed such as NeuN.

the authors said they dissolved 2.5 in water and gave 2.5 mg/kg, so they need how much volume was administered of the solution and what is the concentration. the 2.5mg in water doesn't reflect any concentration. 

Minor:

English writing should be revisited :

many sentences are fragmented: 

for example, it was mentioned  Complete (Roche): should be Complete protease inhibitor

English writing should be revisited  where many sentences are fragmented: 

the authors said "phenotypic changes through skeletal degradation," and they mean cytoskeletal degradation

Round 2

Reviewer 1 Report

Despite the revisions, the manuscript still falls short of the expected quality, particularly in regard to the figures presented. The IHC and Western blot figures exhibit multiple deficiencies in terms of resolution, labeling, and quantification, among other issues. These shortcomings not only make it difficult to interpret the data but also question the scientific rigor of the work presented. Therefore, I cannot recommend this manuscript for publication in its current form.

1. The resolution of all the figures, including IHC and Western blots, is low, making it difficult to interpret the results (especially Figure 1).

2. Several figures lack essential labels, such as markers for axes in the graphs or identification of bands in Western blots. This omission makes it hard to understand the context and significance of the data presented.

3. There is no mention of the number of replicates performed for each experiment, particularly for IHC and Western blotting. This absence raises concerns about the reliability of the data.

4. The scaling in several figures appears inconsistent, which could potentially mislead the reader when interpreting the results.

5. Some figures, particularly the IHC images, seem to contain artifacts that could be mistaken for actual data, thereby affecting the validity of the results.

6. There is a noticeable lack of standardization in the figures, such as the use of different units, fonts, or styles, which makes the manuscript appear unprofessional and hard to follow.

In addition to the issues with the figures, the quality of English in this manuscript needs substantial improvement. The frequent grammatical errors, unclear language, and inconsistencies severely hinder the manuscript's readability and credibility. As such, I cannot recommend this manuscript for publication unless comprehensive language revisions are made.

Reviewer 3 Report

thank you for the corrections

still I can not see the number of animals of animals

Round 3

Reviewer 1 Report

While the Western blot results have shown improvement in quality, there are still significant issues that need to be addressed before this manuscript can be considered for publication in the IJMS.

1. Figure 1 suffers from poor resolution, making it difficult to interpret the data accurately. The quality of this figure needs to be enhanced to meet the journal's standards.

2. There is a concerning discrepancy in Figure 1B. The visual representation suggests that the number of NeuN-positive cells at the epicenter is lower than in the caudal region. However, the statistical results contradict this, indicating that the number of NeuN-positive cells is significantly higher at the epicenter. This inconsistency raises questions about the validity of the data and requires clarification.

3. The manuscript mentions that a SCI model was used but fails to specify which part of the spinal cord was injured. This is a critical detail that needs to be included to assess the study's relevance and applicability.

Due to these issues, I cannot recommend this manuscript for publication in IJMS in its current form.

Author Response

Reviewer’s comments, author responses, and manuscript changes

We thank the referees for carefully reviewing the manuscript and their opinions regarding its scientific content and presentation. In what follows, the reviewers’ comments are in italics, the author’s responses are in blue, and the changes are highlighted in the manuscript.

Reviewer #1

  1. Figure 1 suffers from poor resolution, making it difficult to interpret the data accurately. The quality of this figure needs to be enhanced to meet the journal's standards.

Response: Regarding the resolution of Figure 1, we respectfully ask the referee to review the attached figures file since the resolution of all of them is 300 DPI, as indicated by the journal standards. In addition, we modified Figure 1A to show the histology detail at 100x magnification.

  1. There is a concerning discrepancy in Figure 1B. The visual representation suggests that the number of NeuN-positive cells at the epicenter is lower than in the caudal region. However, the statistical results contradict this, indicating that the number of NeuN-positive cells is significantly higher at the epicenter. This inconsistency raises questions about the validity of the data and requires clarification.

Response: We appreciate the observation that made us realize we made a mistake. To correct this inconsistency, we now manually quantified the NeuN-positive cells and colocalized them with DAPI. We also modified the representative figure and corrected the graph.

  1. The manuscript mentions that a SCI model was used but fails to specify which part of the spinal cord was injured. This is a critical detail that needs to be included to assess the study's relevance and applicability.

Response: The manuscript does mention the specific site of the spinal cord where the injury was performed. This information can be found in the section "4.2 Surgical Procedure" of the Material and Methods section, where we state that SCI was performed at the level of thoracic vertebra 9 (T9). However, to make this information more visible, we modified the wording of that sentence. We also added the information in the Summary and in the legend of Figure 8 (experimental design). The modifications are highlighted in the manuscript.

Round 4

Reviewer 1 Report

Thank you to the authors for their diligent revisions. The quality of the manuscript has significantly improved compared to the first submission. I believe it is now suitable for publication in the IJMS.

Minor editing of the English language required